# Robust Multi-Scale Implicit Neural Representations for Large-Deformation Lung Registration

**Johannes B. Gebauer**[1,2] [iD]                                           J.GEBAUER@UKE.DE

**Maximilian Nielsen**[1,2] [iD]                                            M.NIELSEN@UKE.DE

**Frederic Madesta**[1,2] [iD]                                             F.MADESTA@UKE.DE

**René Werner**[1,2] [iD]                                                   R.WERNER@UKE.DE

**Thilo Sentker**[1,2] [iD]                                                 T.SENTKER@UKE.DE

[1] *Institute for Applied Medical Informatics, University Medical Center Hamburg-Eppendorf, Hamburg, Germany*

[2] *Institute of Computational Neuroscience, University Medical Center Hamburg-Eppendorf, Hamburg, Germany*

**Editors:** Accepted for publication at MIDL 2026

## Abstract

We propose a multi-scale Implicit Neural Representation (INR) framework for dense deformable image registration, designed to stabilize convergence for large deformations while preserving precision for fine anatomical details. We model the INR as a dual-branch architecture that explicitly decomposes the motion into global and local components. The objective function is driven by mask-guided Normalized Cross-Correlation augmented by geometric and semantic regularization to ensure smooth, anatomically plausible motion. Evaluation on the DIR-Lab 4DCT thorax dataset demonstrates competitive performance with a mean Target Registration Error (TRE) below 1.0 mm. On the more challenging DIR-Lab COPDgene thorax dataset, the model achieves robust alignment with a mean TRE of 1.23 mm, yielding performance comparable to leading classical optimization frameworks. A comprehensive ablation study confirms that the dual-branch design and multi-scale optimization strategy are necessary to achieve these results, enabling stable registration with modest computational overhead. Source code is available at https://github.com/IPMI-ICNS-UKE/DUAL-INR-DIR.

**Keywords:** Implicit Neural Representations, Deformable Image Registration, Multi-Scale Optimization, Thoracic CT

## 1. Introduction

Accurate deformable image registration (DIR) is essential for numerous clinical applications, ranging from image-guided interventions and quantitative longitudinal analysis to multimodal image fusion (Sotiras et al., 2013). The challenge lies in estimating a dense, non-rigid transformation field that achieves precise spatial alignment between a moving and a fixed image.

Traditionally, DIR methods, whether based on classical optimization or deep learning (DL) with convolutional neural networks (CNN) (Polzin et al., 2013; Vishnevskiy et al., 2016; Balakrishnan et al., 2019; Hering et al., 2021; Hansen and Heinrich, 2021), rely on representing the deformation field via discrete voxel grids. Consequently, the spatial resolution of the deformation field is inherently tied to the grid density. While interpolation schemes

(e.g., B-splines) provide continuity (Rueckert et al., 2002), capturing finer high-frequency details requires denser control point grids, which increases the number of optimization parameters.

To overcome these structural constraints, we adopt the paradigm of implicit neural representations (INRs). INRs model signals as continuous functions parameterized by a multi-layer perceptron (MLP), offering a resolution-independent mechanism to encode spatial transformations (Molaei et al., 2023). In this framework, the deformation field is modeled as a continuous function of spatial coordinates, where the network predicts the displacement vector. To capture high-frequency components of the deformation that standard MLPs often fail to resolve (spectral bias), we utilize a network with periodic activation functions (SIREN) (Sitzmann et al., 2020), proven to be a powerful choice for DIR as demonstrated by the IDIR framework (Wolterink et al., 2022). An additional advantage of modeling the deformation with an MLP is its inherent differentiablity. This allows the analytic computation of spatial derivatives via automatic differentiation and simplifies the implementation of complex smoothness priors, such as curvature regularization. Finally, unlike supervised DL methods, INRs require no external training data and are analogous to classical iterative schemes in that the deformation field is optimized specifically for a single pair of images.

However, while the standard SIREN approach (Wolterink et al., 2022) is effective at capturing fine details, it often converges to sub-optimal local minima when facing large anatomical motion, resulting in limited registration accuracy. To address this limitation, we advance the INR-based framework by integrating principles from classical registration theory, specifically multi-resolution strategies and spectral decoupling, into a unified multi-scale, dual-branch architecture. Unlike the standard single-branch design, our framework explicitly decouples global anatomical motion from fine-grained local deformations. By optimizing these components in a coarse-to-fine manner, our approach ensures robust alignment for large anatomical changes, such as those seen in COPD patients, while retaining the sub-voxel precision characteristic of SIREN-based INR.

## 2. Method

### 2.1. Problem Formulation

Let $I_f, I_m : \Omega \to \mathbb{R}$ denote the fixed and moving images, respectively, defined on a spatial domain $\Omega \subset \mathbb{R}^3$. Further, let $M : \Omega \to \{0, 1\}$ be the binary masks of relevant structures.

The goal of DIR is to estimate a dense, non-linear transformation field $\phi : \Omega \to \Omega$ that spatially aligns $I_m$ to $I_f$, such that the warped moving image $I_m \circ \phi$ is anatomically consistent with $I_f$. Since estimating $\phi$ solely from image intensity data is an ill-posed problem, we formulate the registration as an energy minimization task combining a masked data fidelity term and a regularization prior:

$$\hat{\phi} = \arg\min_{\phi} \ \mathcal{L}_{\mathrm{sim}}(I_f, I_m \circ \phi; M) + \lambda_{\mathrm{reg}} \mathcal{L}_{\mathrm{reg}}(\phi), \tag{1}$$

where $\mathcal{L}_{\mathrm{sim}}$ quantifies the similarity between the fixed and warped images within the region defined by $M$, $\mathcal{L}_{\mathrm{reg}}$ promotes smoothness and topological regularity in the deformation field, and $\lambda_{\mathrm{reg}} > 0$ is a hyperparameter controlling the trade-off between alignment accuracy and deformation plausibility.

## 2.2. Implicit Neural Representation of the Deformation Field

Unlike traditional approaches that parameterize the deformation field on a discrete voxel grid, we model $\phi$ as a continuous function parameterized by a neural network, referred to as an INR. We define the deformation in a residual form:

$$\phi_\theta(\mathbf{x}) = \mathbf{x} + u_\theta(\mathbf{x}), \tag{2}$$

where $\mathbf{x} \in \Omega$ represents the spatial coordinates and $u_\theta : \Omega \to \mathbb{R}^3$ is the displacement field predicted by an MLP with trainable parameters $\theta$. This residual formulation provides an inductive bias towards the identity transformation, ensuring stable initialization and faster convergence.

Standard MLPs with ReLU activation functions suffer from spectral bias, that is the bias towards learning low frequency signal components (Rahaman et al., 2019). To overcome this and capture fine-grained anatomical details, we adopt the SIREN framework (Sitzmann et al., 2020), which has been successfully applied to medical image registration (Wolterink et al., 2022; Sun et al., 2024). A SIREN network utilizes sinusoidal activation functions and can be expressed as a composition of $n_{\mathrm{L}}$ layers:

$$u_\theta(\mathbf{x}) = \mathbf{W}_{n_{\mathrm{L}}}(\psi_{n_{\mathrm{L}}-1} \circ \psi_{n_{\mathrm{L}}-2} \circ \cdots \circ \psi_0)(\mathbf{x}) + \mathbf{b}_{n_{\mathrm{L}}}, \tag{3}$$

where $\psi_i$ denotes the $i$-th layer of the network. Each layer applies an affine transformation defined by a weight matrix $\mathbf{W}_i$ and a bias vector $\mathbf{b}_i$, followed by a component-wise sine non-linearity:

$$\psi_i(\mathbf{y}) = \sin(\mathbf{W}_i \mathbf{y} + \mathbf{b}_i). \tag{4}$$

A distinct property of INRs is their inherent differentiability, making it possible to compute exact gradients, desired for certain loss terms. Sinusoidal activations allow the computation of higher-order derivatives, required, e.g., for the Hessian, as they are infinitely differentiable ($C^\infty$). This is in contrast to ReLU activations, which are differentiable only once. Furthermore, the sampling strategy for training points is not restricted to a regular grid, allowing for flexible, off-grid optimization.

## 2.3. Optimization and Regularization

Our framework adopts an instance-specific optimization strategy, i.e. we optimize the network parameters $\theta$ directly for a given image pair by defining a loss function analogous to the registration objective defined in Eq. 1:

$$\mathcal{L} = \mathcal{L}_{\mathrm{sim}}(I_f, I_m \circ \phi_\theta) + \mathcal{L}_{\mathrm{reg}}(u_\theta). \tag{5}$$

Rather than masking the loss, we instead restrict the coordinate sampling during training to the respective anatomical mask on which we aim to perform the registration.

For the similarity term $\mathcal{L}_{\mathrm{sim}}$, we employ the Normalized Cross-Correlation (NCC) loss, which is robust to intensity variations between the images.

To ensure smoothness, we utilize a curvature regularizer based on the Laplacian of the displacement field, as proposed in (Fischer and Modersitzki, 2003):

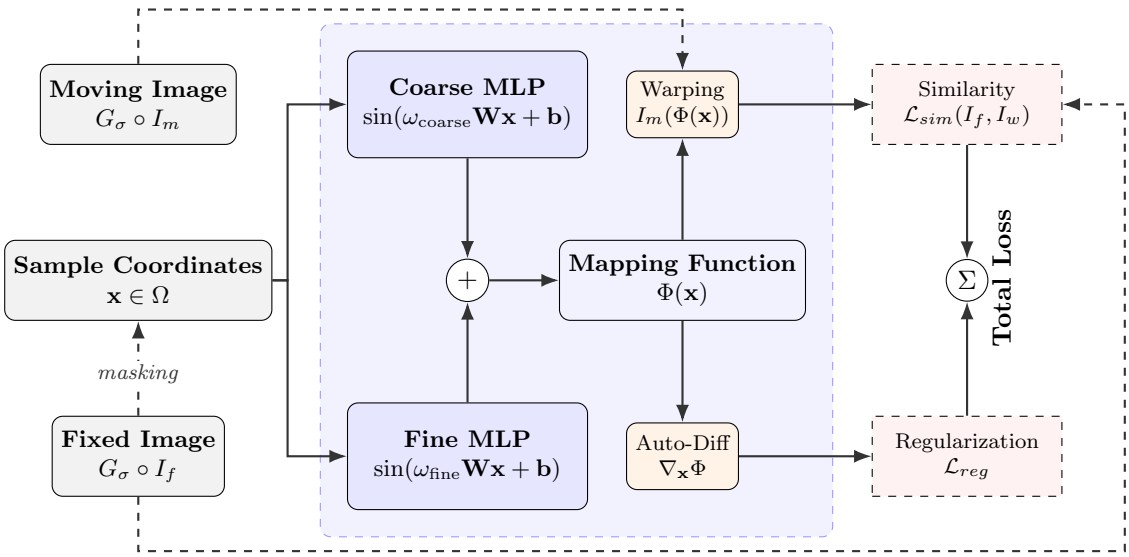

Figure 1: Overview of the proposed multi-scale dual-branch INR framework.

$$\mathcal{L}_{\mathrm{curv}} = \int_{\Omega} \sum_{i=1}^{3} \left( \Delta u_{\theta}^{(i)}(\mathbf{x}) \right)^2 d\mathbf{x}, \tag{6}$$

where $u_{\theta}^{(i)}$ denotes the $i$-th component of the displacement vector.

Furthermore, to prevent folding in the deformation field, we enforce a regularization penalty on negative Jacobian determinants:

$$\mathcal{L}_{\mathrm{jac}} = \int_{\Omega} \mathrm{ReLU} \left( - \left( \det \left( I + \nabla u_{\theta}(\mathbf{x}) \right) - 1 \right) \right) d\mathbf{x} \tag{7}$$

where the ReLU function penalizes only regions with negative Jacobian determinants (indicating local folding), while preserving differentiability.

Finally, to enforce anatomical consistency, we incorporate a mask-based semantic loss. Specifically, we compute the binary cross-entropy between the fixed and warped moving masks for all five lung lobes (left upper/lower; right upper/middle/lower). The resulting composite regularization objective combines smoothness and semantic constraints:

$$\mathcal{L}_{\mathrm{reg}} = \lambda_{\mathrm{curv}} \mathcal{L}_{\mathrm{curv}} + \lambda_{\mathrm{jac}} \mathcal{L}_{\mathrm{jac}} + \lambda_{\mathrm{mask}} \mathcal{L}_{\mathrm{mask}}. \tag{8}$$

We compute all gradients in Eqs. 6 and 7 in a continuous fashion utilizing the autograd functionality in PyTorch (Paszke et al., 2019).

## 2.4. Multi-Scale Dual-Branch Optimization

To effectively recover large deformations, we implement a coarse-to-fine optimization strategy inspired by classical multi-resolution registration techniques. We mitigate the risk of getting trapped in local minima by initially optimizing the network on smoothed image pairs

obtained via Gaussian filtering. The smoothed intensity $G_\sigma$ at voxel coordinates $(i, j, k)$ is defined using a kernel size of $2\sigma + 1$:

$$G_\sigma[i, j, k] = \frac{1}{(2\pi)^{3/2}\sigma^3} \exp\left(-\frac{i^2 + j^2 + k^2}{2\sigma^2}\right), \tag{9}$$

where $\sigma$ controls the smoothing scale. This filtration suppresses high-frequency details, forcing the network to focus on global structural alignment in the early stages. We adopt a multi-stage approach, reducing $\sigma$ in discrete steps ($\sigma \in \{4, 2\}$) as the optimization progresses. In the final stage ($\sigma = 0$), the original non-filtered image is utilized.

To further disentangle global and local motion, we decompose the displacement field into two parallel MLP branches, a coarse branch and a fine branch, as illustrated in Fig. 1. We let both branches have 2 hidden layers with 256 units, respectively. The residual deformation (Eq. 2) is modified to:

$$\phi_\theta(\mathbf{x}) = \mathbf{x} + c_{\text{coarse}} \cdot u_{\theta,\text{coarse}}(\mathbf{x}) + c_{\text{fine}} \cdot u_{\theta,\text{fine}}(\mathbf{x}), \tag{10}$$

where $u_{\theta,\text{coarse}}$ and $u_{\theta,\text{fine}}$ represent the outputs of the respective branches, and $c_{\text{coarse}}, c_{\text{fine}} \in \mathbb{R}$ are additional learnable scaling parameters. The final deformation is therefore the additive combination (residual formulation) of the outputs of the two branches.

An important hyperparameter of SIRENs is the frequency scaling factor $\omega_0$, which effectively scales the spatial frequency of the first network layer and is commonly set to 30 (Sitzmann et al., 2020). To explicitly bias the coarse branch towards global, low-frequency motion, we set $\omega_{\text{coarse}} = 10$, while the fine branch retains the standard high-frequency initialization of $\omega_{\text{fine}} = 30$. Our sensitivity analysis indicates that while the method remains stable for $\omega_{\text{coarse}} \in [10, 30]$, the selected ratio provides the optimal trade-off between global captures and local refinement.

The training procedure is sequential:

1. **Phase 1 (Coarse):** Only the coarse branch is optimized using heavily smoothed images ($\sigma = 4$). The fine branch is inactive ($c_{\text{fine}} = 0$).

2. **Phase 2 (Refinement):** As training progresses, we reduce $\sigma$ to 2 and finally to 0. Simultaneously, we activate the fine branch. To ensure a stable transition, $c_{\text{fine}}$ is initialized at a lower magnitude relative to the coarse scale (specifically $c_{\text{fine}} = c_{\text{coarse}}/10$), allowing the network to progressively add local details to the established global deformation.

We employ this multi-step training scheme because we observed that the fine branch tends to collapse to zero if active from the start.

### 2.5. Implementation Details

We optimize the network by iteratively sampling random coordinates $\mathbf{x}$ strictly from within the fixed image lung mask $M_f$. Lung and lung lobe masks are automatically generated with TotalSegmentator (Wasserthal et al., 2023). At each iteration, we sample a batch of $N = 30,000$ points. We train for a cumulative total of 3,000 steps distributed across the three optimization stages ($\sigma \in \{4, 2, 0\}$). We employ the AdamW optimizer with an

initial learning rate of $10^{-3}$, which is gradually reduced to $10^{-5}$ following a cosine annealing schedule.

For the network topology, we utilize the dual-branch architecture with layer dimensions and frequency initializations ($\omega_0$) as defined in Sec. 2.4. We additionally follow the proposed SIREN-initialization scheme and draw the initial network weights form the uniform distribution $\mathbf{W}_i \sim \mathcal{U}(-\sqrt{6/n}, \sqrt{6/n})$, where $n$ is the number of inputs to the layer. The scaling factors are initialized as $c_{\text{coarse}} = 0.1$ and, at the point where the fine branch is activated, $c_{\text{fine}} = 0.05 \cdot c_{\text{coarse}}$. The loss weighting hyperparameters are set to $\lambda_{\text{curv}} = 0.01$, $\lambda_{\text{jac}} = 0.01$, and $\lambda_{\text{mask}} = 0.1$.

All experiments were conducted on a single NVIDIA 3090 GPU using PyTorch. All hyperparameters are kept constant across all cases to ensure a strict evaluation of the method's generalization capabilities over diverse anatomical deformations.

## 2.6. Datasets

We evaluate the proposed INR-based registration framework using two standard benchmarks for thoracic image registration: the DIR-Lab 4D Computed Tomography (4DCT) dataset (Castillo et al., 2009) and the DIR-Lab COPDgene dataset (Castillo et al., 2013). Both datasets include manually annotated anatomical landmarks, enabling the quantitative assessment of Target Registration Error (TRE).

### 2.6.1. DIR-Lab 4DCT

The DIR-Lab 4DCT dataset comprises 10 subject-specific 4DCT scans. Each case contains 10 volumetric CT images representing distinct phases of the respiratory cycle (phases 0 through 9), spanning from maximum inhalation to maximum exhalation. To evaluate registration accuracy, we utilize the provided set of 300 expert-identified landmarks for each case, which are defined on the extreme phases (end-inspiration and end-expiration).

The image dimensions and voxel spacings vary across subjects:

- **Cases 1–5:** Spatial resolution of $256 \times 256 \times [99\text{–}112]$ voxels, with anisotropic voxel spacing ranging from $[0.97\text{–}1.16] \times [0.97\text{–}1.16] \times 2.5\,\text{mm}^3$.

- **Cases 6–10:** Higher spatial resolution of $512 \times 512 \times [106\text{–}136]$ voxels, with a fixed voxel spacing of $0.97 \times 0.97 \times 2.5\,\text{mm}^3$.

### 2.6.2. DIR-Lab COPDgene

To assess performance on pathology-induced large deformations, we utilize the DIR-Lab COPDgene dataset. This set consists of 10 image pairs derived from patients with Chronic Obstructive Pulmonary Disease (COPD), providing distinct volumetric breath-hold CT images for the inhalation (iBH) and exhalation (eBH) phases.

The spatial parameters for this dataset are grouped as follows:

- **Case 2:** Spatial resolution of $256 \times 256 \times 112$ voxels, with a physical spacing of $0.645 \times 0.645 \times 2.5\,\text{mm}^3$.

- **Cases 1 & 3–10:** Spatial resolution of $512 \times 512 \times [112\text{–}135]$ voxels, with in-plane spacing ranging from $0.586$ to $0.742\,\text{mm}$ and a fixed slice thickness of $2.5\,\text{mm}$.

## 2.7. Evaluation Metrics

To quantitatively assess registration performance, we utilize three standard metrics focusing on landmark accuracy, anatomical overlap, and deformation regularity.

The primary metric for registration accuracy is the Target Registration Error (TRE). We follow the standard evaluation protocol for the DIR-Lab benchmarks. For each corresponding landmark pair $(\mathbf{p}_m, \mathbf{p}_f)$ in the moving and fixed domains, the fixed landmark is first transformed by the estimated deformation field to obtain $\mathbf{p}'_f = \phi(\mathbf{p}_f)$. This transformed coordinate is then snapped (rounded) to the nearest integer voxel index. The final TRE is computed as the Euclidean distance between this snapped coordinate and the moving landmark $\mathbf{p}_m$ in physical world coordinates (millimeters). We report the mean TRE and standard deviation over all 300 expert landmarks provided for each case.

To evaluate the spatial overlap of anatomical structures, we compute the Dice Similarity Coefficient (DSC). Since our registration framework focuses specifically on the pulmonary region, we assess alignment accuracy using the segmentations of the five anatomical lung lobes described in Sec. 2.3.

## 2.8. Experiments

In addition to the evaluation of our proposed method (see Sec. 2.5), we conducted the following experiments to assess the contributions of components and the robustness.

To systematically evaluate the individual contributions of the proposed dual-branch architecture and the multi-scale optimization schedule relative to standard INR implementations, we perform an ablation study across both datasets using the following configurations:

1. **Single-Branch:** A standard SIREN network (3 hidden layers, 256 units, $\omega_0 = 30$) trained directly on high-resolution images, representing the vanilla INR approach.

2. **Single-Branch + Multi-scale:** The standard SIREN trained using our coarse-to-fine Gaussian smoothing schedule, testing the impact of scheduled learning on a standard architecture.

3. **Dual-Branch:** Our proposed dual-branch architecture trained directly on high-resolution images without the smoothing schedule, testing the capability of the architecture alone to decouple motion.

4. **Dual-Branch + Multi-scale (Full Model):** The complete proposed framework, combining the dual-branch architecture with the coarse-to-fine Gaussian smoothing schedule.

Prior work has shown that INRs can be sensitive to weight initialization (Harten et al., 2023). To demonstrate the robustness of our method, all ablation configurations were trained on 10 distinct random seeds.

Further, to evaluate the influence of the individual loss components, we performed an ablation study of the regularization terms defined in Eq. 8. We applied this analysis to both the single-branch baseline (configuration 1) and our proposed method (configuration 4). We restrict this to the COPDgene dataset, as this contains the more challenging cases.

## 3. Results

We evaluated our dual-branch INR framework on all 20 cases described in Sec. 2.6. In Table 1, the resulting TRE values are compared against one classical registration approach (isoPTV) and competing DL-based methods. For IDIR (Wolterink et al., 2022), we report values obtained using the official implementation with default hyperparameters, as case-wise results for COPDgene were not available in the original publication. On average, our proposed model outperforms all learning-based models across both datasets. While it does not fully match the precision of the classical isoPTV benchmark, the performance gap compared to previous INR approaches is reduced.

While performance is comparable to IDIR on cases with small initial displacements (i.e., 4DCT cases 1 and 2), our model demonstrates superior robustness in scenarios with large anatomical deformations, particularly within the COPDgene cohort.

Further inspection of the individual network branches confirms that our training scheme effectively separates global and local motion. The coarse branch captures the majority of the deformation, with mean displacement magnitudes $6\times$ and $14\times$ larger than those of the fine branch for the 4DCT and COPDgene datasets, respectively. This decomposition is visually exemplified in Fig. 2. Additional qualitative results, i.e. pairs of fixed, moving and warped images, are shown in Fig. 4.

### 3.1. Ablation Study

The quantitative results of the architectural ablation study over 10 different random seeds are shown in Table 2 and Fig. 3. On the 4DCT dataset, performance differences between the single-branch model variants and our proposed method were negligible. However, the dual-branch without multi-scale training showed instability, failing to converge in four out

Table 1: Quantitative comparison on DIR-Lab 4DCT and DIR-Lab COPDgene data (Cases 1–10), evaluated by TRE (mean and std) in mm for each individual case, along with the mean performance over all cases. We compare our method against one classical approach (isoPTV) and other learning-based approaches: GraphRegNet (deep graphs), VIRNet (CNN), IDIR (INR). Bold indicates the best result.

| Method | DIR-Lab 4DCT Case ID | | | | | | | | | | |
| | 1 | 2 | 3 | 4 | 5 | 6 | 7 | 8 | 9 | 10 | Overall |
| --- | --- | --- | --- | --- | --- | --- | --- | --- | --- | --- | --- |
| no reg. | 4.01 (2.91) | 4.65 (4.09) | 6.73 (4.21) | 9.42 (4.81) | 7.10 (5.15) | 11.10 (6.98) | 11.59 (7.87) | 15.16 (9.11) | 7.82 (3.99) | 7.63 (6.54) | 8.52 (5.57) |
| isoPTV[1] | **0.76 (0.90)** | 0.77 (0.89) | 0.90 (1.05) | **1.24 (1.29)** | **1.12 (1.44)** | **0.85 (0.89)** | **0.80 (1.28)** | 1.34 (1.93) | **0.92 (0.94)** | **0.82 (0.89)** | **0.95 (1.15)** |
| GraphRegNet[2] | 0.86 (N/A) | 0.90 (N/A) | 1.06 (N/A) | 1.45 (N/A) | 1.60 (N/A) | 1.59 (N/A) | 1.74 (N/A) | 1.46 (N/A) | 1.58 (N/A) | 1.71 (N/A) | 1.39 (N/A) |
| VIRNet[3] | 0.99 (0.47) | 0.98 (0.46) | 1.11 (0.61) | 1.37 (1.03) | 1.32 (1.36) | 1.15 (1.12) | 1.05 (0.81) | 1.22 (1.44) | 1.11 (0.66) | 1.05 (0.72) | 1.14 (0.76) |
| IDIR[4] | **0.76 (0.94)** | **0.76 (0.94)** | 0.94 (1.02) | 1.32 (1.27) | 1.23 (1.47) | 1.09 (1.03) | 1.12 (1.00) | 1.21 (1.29) | 1.22 (0.95) | 1.01 (1.05) | 1.07 (1.10) |
| Ours | **0.76 (0.91)** | **0.76 (0.91)** | **0.88 (1.03)** | 1.27 (1.22) | 1.16 (1.46) | 0.95 (0.99) | 0.93 (0.97) | **1.10 (1.25)** | 1.00 (0.93) | 0.94 (0.95) | 0.98 (1.07) |

| | DIR-Lab COPDgene Case ID | | | | | | | | | | |
| | 1 | 2 | 3 | 4 | 5 | 6 | 7 | 8 | 9 | 10 | Overall |
| --- | --- | --- | --- | --- | --- | --- | --- | --- | --- | --- | --- |
| no reg. | 25.90 (11.57) | 21.77 (6.46) | 12.29 (6.39) | 30.90 (13.49) | 30.90 (14.05) | 28.32 (9.20) | 21.66 (7.66) | 25.57 (13.61) | 14.84 (10.01) | 22.48 (10.64) | 23.46 (10.31) |
| isoPTV[1] | **0.77 (0.75)** | 2.22 (2.94) | **0.82 (0.80)** | **0.85 (0.86)** | **0.77 (0.84)** | **0.86 (1.92)** | **0.74 (1.06)** | **0.81 (1.84)** | **0.83 (1.22)** | **0.92 (0.85)** | **0.96 (1.31)** |
| GraphRegNet[2] | 1.38 (N/A) | 2.09 (N/A) | 1.22 (N/A) | 1.58 (N/A) | 1.37 (N/A) | 1.10 (N/A) | 1.19 (N/A) | 1.19 (N/A) | 0.99 (N/A) | 1.38 (N/A) | 1.34 (N/A) |
| VIRNet[3] | — | — | — | — | — | — | — | — | — | — | — |
| IDIR[4] | 1.55 (1.67) | 2.66 (3.57) | 1.35 (1.07) | 1.46 (1.18) | 1.43 (1.54) | 66.46 (31.87) | 1.39 (1.33) | 1.72 (2.00) | 1.28 (1.50) | 1.62 (1.28) | 8.09 (4.70) |
| Ours | 1.11 (1.26) | **2.07 (3.19)** | 1.11 (0.98) | 1.10 (0.94) | 1.09 (1.16) | 1.40 (2.07) | 1.04 (1.19) | 1.27 (1.68) | 0.95 (1.20) | 1.21 (1.14) | 1.23 (1.48) |

[1](Vishnevskiy et al., 2016) [2](Hansen and Heinrich, 2021) [3](Hering et al., 2021) [4](Wolterink et al., 2022)

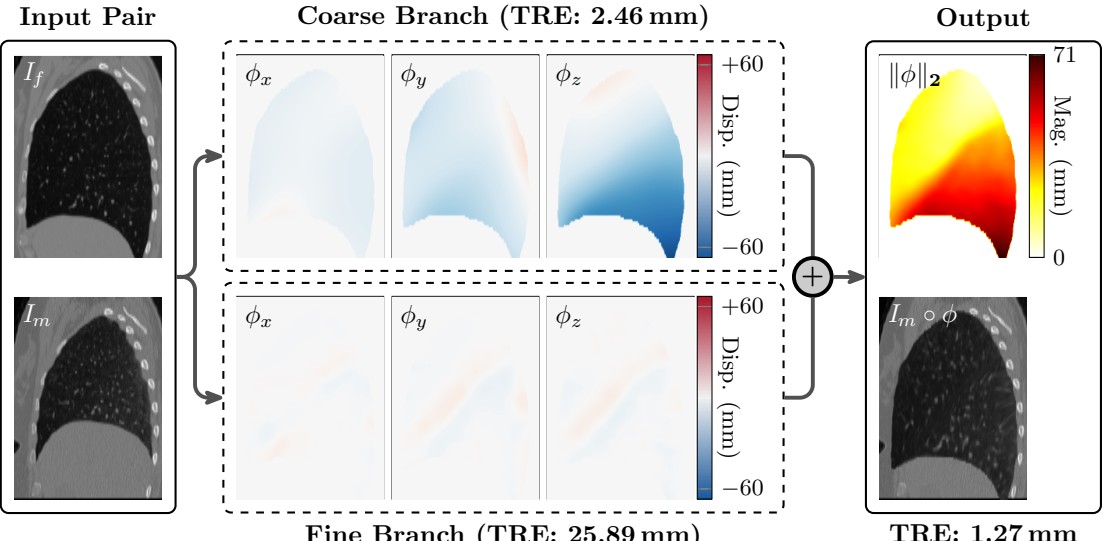

Figure 2: Visual decomposition of the registration field (Case 8, DIR-Lab COPDgene). Left: Fixed ($I_f$) and moving ($I_m$) input images. Center: The learned displacement fields are decomposed into $x$ (lateral), $y$ (anterior-posterior), and $z$ (inferior-superior) components ($\phi_x, \phi_y, \phi_z$) for both the coarse and fine branches. Right: The magnitude field $||\phi||_2$ and the resulting warped image $I_m \circ \phi$.

Table 2: Ablation Study on DIR-Lab 4DCT and COPDgene datasets. We evaluate the impact of the dual-branch architecture and multi-scale (MS) optimization on registration accuracy (TRE), overlap (DSC), and topological regularity ($|J_\phi| \leq 0$). Values represent mean $\pm$ standard deviation over 10 random seeds.

| Configuration | | DIR-Lab 4DCT | | | DIR-Lab COPD | | |
|---|---|---|---|---|---|---|---|
| Arch. | MS | TRE (mm) $\downarrow$ | DSC $\uparrow$ | $|J_\phi| \leq 0$ (%) $\downarrow$ | TRE (mm) $\downarrow$ | DSC $\uparrow$ | $|J_\phi| \leq 0$ (%) $\downarrow$ |
| SINGLE | $-$ | $1.02 \pm 0.19$ | $0.964 \pm 0.009$ | $0.001 \pm 0.002$ | $1.42 \pm 0.36$ | $0.952 \pm 0.010$ | $0.026 \pm 0.038$ |
| SINGLE | $\checkmark$ | $1.02 \pm 0.19$ | $0.964 \pm 0.009$ | $0.001 \pm 0.001$ | $1.43 \pm 0.37$ | $0.952 \pm 0.010$ | $0.025 \pm 0.035$ |
| DUAL | $-$ | $1.35 \pm 2.00$ | $0.964 \pm 0.012$ | $0.107 \pm 0.606$ | $2.64 \pm 2.70$ | $0.948 \pm 0.020$ | $0.721 \pm 1.147$ |
| DUAL | $\checkmark$ | $0.97 \pm 0.15$ | $0.964 \pm 0.009$ | $0.000 \pm 0.000$ | $1.21 \pm 0.29$ | $0.952 \pm 0.011$ | $0.151 \pm 0.235$ |

of ten runs for case 9. On the more complex COPDgene dataset, our proposed model demonstrated a performance gain over the single-branch architectures. Consistent with the 4DCT findings, the dual-branch without multi-scale guidance failed to converge in six out of ten cases. Employing the multi-scale schedule for the single-branch network shows no performance change, which confirms that the schedule is specifically required to stabilize the higher capacity dual-branch design. Quantitative analysis of deformation regularity of our proposed method reveals that negative Jacobian determinants occur in 0.15% of the voxel locations on average for the COPDgene and never for the 4DCT data. This

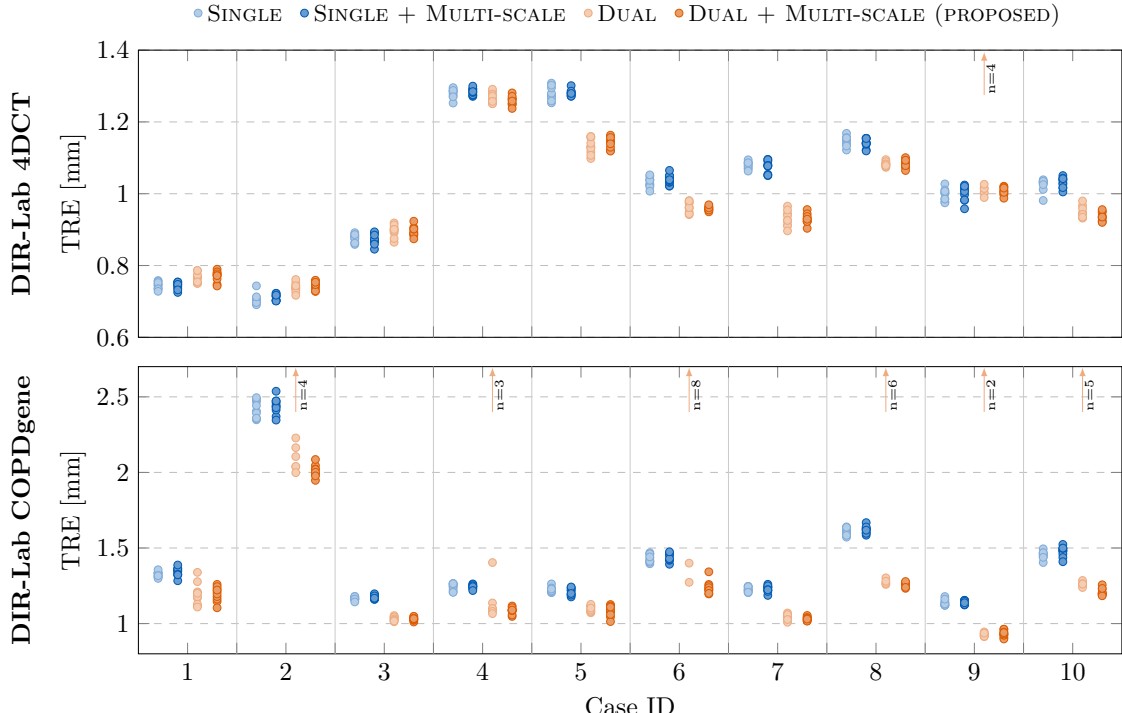

Figure 3: Robustness analysis across 10 random seeds for the DIR-Lab 4DCT (top) and COPDgene (bottom) datasets. The stability and accuracy of single-branch and dual-branch architectures, with and without the proposed multi-scale optimization schedule, are compared using TRE. Arrows (e.g., n=4) indicate the number of divergent runs where the TRE exceeded the plot range (catastrophic failure), observed solely in the dual-branch configuration without multi-scale guidance.

indicates that the method maintains high topological plausibility and prevents folding in the majority of the anatomical domain. The single-branch architectures yield even smoother displacement fields for the COPDgene data, likely due to their limited capacity to model complex deformations, as shown by their lower registration accuracy. Finally, DSC scores remained consistent across all ablation models, indicating that even in cases with high TRE (divergence), the model successfully aligned the global lung boundaries, suggesting that the instability primarily manifests as distortions in the internal deformation field.

The loss function ablation on the DIR-Lab COPDgene dataset demonstrates that while all regularization components contribute to the final performance, curvature regularization is the most critical individual term. Using curvature regularization alone yields performance almost as high as the combined loss function (mean TRE of 1.25 mm); however, the combination of all three terms still achieves the best overall balance of accuracy and topological regularity. Optimization using only the mask loss yields the poorest results. The dual-branch architecture remains robust even in this setting, maintaining a mean TRE

**DIR-Lab COPDgene Dataset**

Figure 4: For each case (1–5), the top row displays the fixed image overlaid with a color-coded (color coding is representing motion in lateral direction) static reference grid, while the middle row illustrates the moving image with the grid warped according to the computed displacement vector field. The bottom row presents the correspondingly warped moving image.

of 2.09 mm. In contrast, the single-branch baseline fails without geometric regularization (mean TRE > 25 mm). Finally, without any regularization, the dual-branch architecture yields a mean TRE of ∼6 mm, whereas the single-branch baseline diverges to ∼32 mm.

## 4. Discussion

In this work, we proposed a robust INR framework for DIR, specifically designed to handle large anatomical deformations. Inspired by classical DIR methods, we introduced a multi-scale optimization schedule paired with a dual-branch architecture to decouple global and local motion components. Our evaluation on the DIR-Lab 4DCT and COPDgene benchmarks confirms the efficacy of this approach, yielding competitive precision on standard respiratory motion (4DCT) and robust performance on pathology-induced (COPDgene) deformations.

Our ablation studies demonstrate that decoupling motion via the dual-branch architecture yields modest improvements for small deformations but considerable performance gains for large-scale motion compared to its single-branch counterpart. However, the multi-scale

training scheme is essential for the stability of this approach. Without it, the increased flexibility of the dual-branch model leads to frequent divergence in complex cases. Our intuition is that without the spectral guidance provided by the smoothing schedule, the two branches compete to model overlapping frequency bands. This creates an optimization landscape where the coarse branch fails to map global motion, causing the optimization to become trapped in local minima. The multi-scale schedule prevents this by forcing the coarse branch to converge on the global structure before high-frequency details are introduced. Finally, the robustness of our full method is confirmed by the low variance in mean TRE values across different random network initializations, indicating that it reliably converges regardless of the starting seed.

Despite these promising results, our approach has limitations inherent to instance-specific INRs. First, the computational cost exceeds that of single-shot learning methods. In our current implementation, optimizing a single case requires 5–10 minutes. However, we observed that registration metrics often stabilize within the first 60 seconds, especially for smaller deformations. Consequently, future work focused on finding appropriate and reliable stopping criteria could accelerate the optimization process. Second, although robust to large deformations, modeling physiological sliding motion (e.g., at the pleural boundary) remains challenging due to the continuous nature of vector fields. Future work could investigate spatially adaptive regularization and specialized sampling schemes to better resolve these interfaces.

Overall, we have presented a robust, coordinate-based registration framework for thoracic 4DCT. By effectively combining the continuous representation power of SIREN in a dual-branch architecture with a classical coarse-to-fine optimization strategy, our method offers a powerful tool for medical image analysis where precise alignment is highly relevant.

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
