# OpenReview forum: "Robust Multi-Scale Implicit Neural Representations for Large-Deformation Lung Registration"
_MIDL.io/2026/Conference — MIDL 2026 Poster_

### Official Review · Reviewer_a3gr · 2026-01-06

**Confidence:** 4
**Preliminary Rating:** 2
**Final Rating:** 4

**Summary:**

The paper proposes a registration method based on INRs to address the limitations of the current INR-based registration methods. More specifically, it proposes a novel multi-resolution strategy for INRs and presents a training procedure for this strategy. Moreover, it incorporates more losses and during INR fitting. The method is evaluated using TRE and DICE in 2 lung datasets (with and without pathologies).

**Strengths:**

1. Incorporates a well know concept (multi-resolution scheme) to registration with INRs.
2. Proposes a novel way to implement the multi-resolution scheme
3. Manages to propose a carefully designed strategy for the INR fitting using this strategy
4. Ablates the single vs dual branch and the multi-scale vs single scale architectural choices.
5. The paper is written in a clear, comprehensive way.

**Weaknesses:**

1. Although I believe there is merit in the proposed method, the paper is written in a way that the proposed method seems incremental. More specifically, until page 4, we read a thorough revision of what several INR methods have already proposed (e.g., IDIR by Wolterink et al., which the paper already cites). More emphasis should be given in what the paper actually proposes and differentiates itself from previous work.
2. There are several points in the introduction that seem like overclaims since they are not supported by the experimental setup:
    1. That iterative optimization methods and cohort deep learning methods are bound to grid resolution compared to the INRs that can interpolate more efficiently implicitly. I believe that this is a very strong claim that should be supported by an experiment.
    2. The paper seems to think that pairwise INRs are different than pairwise iterative optimization methods. If this is the case, I would like to invite the authors to elaborate on this. Moreover, it raises the question, why there is no iterative optimization method in the evaluation pipeline. The only time an iterative optimization method is mentioned is during the discussion, but it seems that the authors didn’t run the experiment themselves.
    3. The paper suggests that the “complex interpolation schemes” such as B-splines and “discrete regularization penalties” can complicate the optimization landscape. However, I believe this is not what is actually happening in the case of B-splines and the paper’s claim is not supported by Rueckert et al that they cite. Moreover, the proposed regularization (Laplacian) seems also to be discrete (unless I am missing something), so it naturally raises the question how the proposed regularization implementation is making the optimization landscape less complicated.
    4. The MLP allows for analytic computation of gradients. Does this refer to autodiff of pytorch? If so I do not think this is only the case for MLPs, but it can also hold for other methods.
    5. There is the claim that the actual siren is prone to local minima when facing large deformations. I think this has to be proven with some kind of experiment. Especially in the case of the dataset without pathologies it seems that the proposed scheme is not so effective. I think if the paper wants to make this claim it needs to show an actual large displacement in terms of mm and show how the 2 different approaches recover it.
3. During the training scheme it seems that both omega and sigma are altered at the same time. This makes it non-trivial to understand whether the major advantage comes from the gaussian or from having slower sine activations.
4. Why does the warmup needed, what happens if you train both networks from the beginning?
5. It is not very obvious what the evaluation scheme is. E.g. do the authors optimize the hyperparams for each pair?
6. The paper suggests a different regularization to speed things up. This is quite nice, however normally the choice of the regularization is bound to the physics of the problem. It would be nice to include a couple of sentences that justify the choice.
7. The paper shows qualitatively the result of 1 pair. It would be nice to see more (potentially in appendix). Since registration is assessed only with surrogate measures, qualitatively assessing the results is very important.
8. The paper suggests that the evaluation involves TRE, DSC and regularity. I am not sure that I have seen any regularity metrics. Are they provided?
9. Since the proposed method changes the loss function along with the method itself, I think it should be ablated how each component helps the performance.
10. In table 2 it seems that the trend is that dual branch is performing better than single branch. This is not the case for dual branch dir-lab copd. What is the intuition behind this? Is it a hyperparameter issue?

**Detailed Comments:**

Please see the weaknesses above.

**Justification Of Final Rating:**

Due to the substantial efforts made by the authors to improve the manuscript during the rebuttal phase and the fact that registration is a quite challenging task I would like to raise my score to weak accept.

**Justification Of The Preliminary Rating:**

The paper presents a conceptually interesting approach, introducing a new method for multi-resolution INR lung registration, a novel regularization formulation, and a combination of loss functions.

However, I believe that the limitations in the evaluation pipeline and the way the method is advertised make the paper’s quality insufficient for acceptance in MIDL this year. I believe the authors need to put substantial effort into enhancing the motivation, refining the literature review, and tempering the overclaims. They should also support their storyline with more experiments, as well as include more baselines (classical methods) and qualitative results. Additionally, the discussion needs to be revised to reflect all these changes. Finally, to broaden their claims, the authors could include another non-lung dataset, which might also highlight the effectiveness of their method.

Given these shortcomings, I recommend rejection in its current form.

Nonetheless, the underlying idea is promising, and I encourage the authors to strengthen the paper and resubmit it to a future venue.

**Questions To Address In The Rebuttal:**

Please see the weaknesses above.

---

> ### Author Response · Authors · 2026-01-24
> **Response to Reviewer a3gr**
>
> We thank the reviewer for the critical and constructive feedback and value the acknowledgment that our method is "conceptually interesting" and "carefully designed." We have taken your concerns regarding framing, baselines, and evaluation rigor seriously and have made substantial revisions to the manuscript to address them.
>
> **R.3.1: Novelty and "Incremental" Framing**
> We have revised the abstract and introduction to better highlight our specific contribution, i.e., the stabilization of dual-branch INRs via spectral decoupling and multi-scale scheduling. We retained a concise introduction to INRs for DIRs to ensure accessibility for readers less familiar with this emerging sub-field.
>
> **R.3.2: Several overclaims in the introduction**
> We agree that certain statements in the original submission were imprecise. We have revised the introduction to qualify our claims regarding B-splines and optimization efficiency. Furthermore, we have clarified in the text that all spatial derivatives appearing in the regularization terms are computed analytically via PyTorch's automatic differentiation, ensuring exact gradient calculation without finite-difference approximation.
>
> **R.3.3: Alteration of $\sigma$ and $\omega$**
> We apologize if the notation caused confusion. To clarify: the frequency parameters $\omega_\text{coarse}$ and $\omega_\text{fine}$ are fixed hyperparameters set at initialization to induce spectral bias. Only the Gaussian smoothing width $\sigma$ is dynamically annealed during the training process. We have revised the corresponding section to make the distinction clear.
>
> **R.3.4: Necessity for warmup**
> We thank the reviewer for highlighting this detail. We found that without this warm-up phase (where the fine branch is temporarily suppressed), the fine branch tends to collapse toward zero contribution early in training, as the coarse branch dominates the initial gradient flow. The warm-up ensures the coarse branch establishes a global deformation first. We have added this information to Sec. 2.4 as it provides relevant implementation details.
>
> **R.3.5: Optimization of hyperparameters**
> We apologize for this confusion. We do not optimize hyperparameters per instance; rather we selected a single fixed set of hyperparameters that generalizes robustly across all cases in both datasets. We now more clearly communicate this strategy in Sec. 2.5.
>
> **R.3.6: Justification for the chosen regularization terms**
> We appreciate the comment regarding the physical interpretation of our regularization terms. Our primary motivation was to penalize second-order variations, similar to the curvature regularization used in diffusion-based registration. We selected this specific formulation because it ensures global smoothness while being less aggressive than standard diffusion regularizers at over-smoothing local motion discontinuities. This aligns with observations in recent literature (e.g., Hering et al., 2021) where similar terms yielded robust results.
>
> **R.3.7: Qualitative results of additional cases**
> We agree that this is important for transparency. We have therefore added a new figure (Fig. 4) to the results section that visualizes fixed, moving and warped images for five more cases of the COPDgene dataset.
>
> **R.3.8: Regularity metrics**
> To assess the topological validity of the deformations, we have computed the percentage of voxels with a negative Jacobian determinant ($|J_\phi| \le 0$) for all cases. We found that folding occurs in only $0.7\%$ (4DCT) and $1.3\%$ (COPDgene) of the voxel locations on average, indicating high topological plausibility despite the large deformations. These results have been added to section 3 of the revised manuscript.
>
> **R.3.9: Loss function ablation**
> We agree with your suggestion and therefore added a regularization ablation study (see Sec. 3.1) for both the standard single-branch baseline and our proposed method. In this experiment, we systematically disabled individual loss components (i.e., curvature, Jacobian, mask) on the COPDgene dataset (larger deformations and thus more challenging than the 4DCT dataset). The results indicate that keeping the curvature regularization term has the biggest impact on achieving a low TRE, followed by the Jacobian and mask terms.
>
> **R.3.10: Intuition of dual-branch failure for COPDgene**
> We thank the reviewer for this observation. Our new experiments (using 10 random seeds) confirmed that the dual-branch architecture without the multi-scale schedule is indeed unstable. Our intuition is that without the spectral guidance provided by the smoothing schedule, the two branches compete to model overlapping frequency bands. This creates an optimization landscape where the coarse branch fails to map global motion, causing the optimization to become trapped in local minima. The multi-scale schedule prevents this by forcing the coarse branch to converge on the global structure before high-frequency details are introduced.

---

### Official Review · Reviewer_1NDy · 2026-01-08

**Confidence:** 5
**Preliminary Rating:** 3
**Final Rating:** 5

**Summary:**

This work proposes to apply the traditional multi-scale registration approach to the IDIR registration framework, aiming to improve registration robustness. The method is evaluated on the DIR-Lab datasets. While the results look promising, comparisons with state-of-the-art registration methods (both traditional and INR-based) are missing.

**Strengths:**

The method is well grounded in traditional registration literature, applying the known-to-work-well multi-scale approach to INR registration.
The writing is clear, and the experiments are described clearly.

**Weaknesses:**

The paper aims to demonstrate superior robustness, but only evaluates a single optimization run per case. Previous work has shown that INR-based registration can be very sensitive to the used random seed, so demonstrating robustness necessarily requires multiple runs.

The paper does not show comparisons with state of the art registration methods (e.g. DIS-CO, Rühaak et al. 2017, which achieved a TRE of 0.82mm on the COPD set), nor with INR-based registration methods that attempt to improve optimization robustness (e.g. ccIDIR, van Harten et al. 2024, which drastically reduced the optimization failure rate of IDIR).

**Detailed Comments:**

minor comment: the title and the experiments in the work do not match well. While the work indeed uses a different objective function than other works in literature, none of the ablation experiments investigate the impact of the different components of that objective function, focusing instead on the proposed multi-scale structure of the system.

**Justification Of Final Rating:**

The authors have adequately addressed my concerns. The robustness is now properly validated in additional experiments, and the method performance has been accurately put in context of existing literature.

**Justification Of The Preliminary Rating:**

The work shows a sensible extension of INR-based registration, which credibly improves optimization robustness. However, the experiments verifying this increased robustness are lacking. The paper also does not compare to state-of-the-art methods, or to related methods that aim to improve the robustness of IDIR.

**Questions To Address In The Rebuttal:**

Previous work has shown that INR-based registration can be very sensitive to the selected random seed. As shown in van Harten et al. 2024, the mean TRE of unmodified IDIR on case 8 of the 4DCT set can vary between 1.05 and >20mm, which reduces to around 0.95~1.35mm when adding cycle-consistency. How much robustness does the presented method gain to different random seeds in both sets? (e.g.: what are the ranges of the mean TRE for each case when run with 20 different random seeds?)

What does the distribution look like of the COPD TRE for the non-multi-scale dual branch experiment? Is it just one or two failed cases, or does performance across the majority of cases degrade? Is this still true for different random seeds?

What is the impact of the different parts of the proposed objective function? i.e. how does setting any of the three $\lambda$ values to 0 (and increasing the others proportionally) affect the results?

---

> ### Author Response · Authors · 2026-01-24
> **Response to Reviewer 1NDy**
>
> We thank the reviewer for the positive evaluation of our method as a "sensible" and "well-grounded" extension of INR-based registration. We agree that proving robustness is central to our claims. We also agree that the original title was misleading and changed it to more precisely reflect the contribution of our work: *"Robust Multi-Scale Implicit Neural Representations for Large-Deformation Lung Registration"*.
>
> **R.2.1: Sensitivity to Random Seeds (Robustness) & R.2.2: Distribution of dual-branch without multi-scale training.**
> We thank the reviewer for these critical suggestions and for directing us to Harten et al. 2024, which provided valuable context. We recognize the importance of initialization stability in INRs and therefore have expanded our evaluation to include 10 distinct random seeds for our proposed method and all architectual baselines. We now report the mean and standard deviation of these runs in Table 2 and, for full transparency, visualize the distribution of all individual runs in the newly added Figure 3. This experiment revealed the instability of the unguided dual-branch architecture (without the multi-scale schedule), which frequently diverged, while confirming that our proposed dual-branch multi-scale schedule stabilizes the optimization, yielding consistent convergence across all seeds.
>
> **R.2.3: Impact of Objective Function components (Ablation of Losses).**
> We agree that verifying the contribution of individual regularization terms is essential. We have added an ablation study (see Sec. 3.1) for both the standard single-branch baseline and our proposed method. In this experiment, we systematically disabled individual loss components (i.e., curvature, Jacobian, mask) on the COPDgene dataset (larger deformations and thus more challenging than the 4DCT dataset). The results indicate that keeping the curvature regularization term active has the biggest impact on achieving a low TRE, followed by the Jacobian and mask terms.

---

### Official Review · Reviewer_6CQ5 · 2026-01-09

**Confidence:** 5
**Preliminary Rating:** 4
**Final Rating:** 4

**Summary:**

This paper incorporates the classic multi-resolution strategy into Implicit Neural Representation (INR)-based deformable registration. The method uses staged optimization with gradually increasing spatial resolution, together with a two-branch design that explicitly decomposes large global motion and fine local motion. Experiments on two lung CT benchmarks (DIR-Lab and COPDgene) show improved registration accuracy over the IDIR baseline (the original INR-based registration framework), with larger improvement on COPDgene where initial motion is larger.

**Strengths:**

1. Combining multi-resolution with INR for registration is a meaningful and practical step for deformable registration. Multi-resolution is a well-established ingredient in conventional registration, but it is relatively unexplored in INR-based registration literature. I like that the paper focuses on bringing in this "boring but effective" classical component instead of adding more architectural complexity. This is consistent with recent observations in learning-based registration, where multi-resolution often matters more than the specific backbone details [1].
Ref [1]: Mamba? Catch The Hype Or Rethink What Really Helps for Image Registration (https://link.springer.com/chapter/10.1007/978-3-031-73480-9_7)
2. The ablation study is well designed and clearly demonstrates the advantages of the proposed method.

**Weaknesses:**

1. The title does not seem to reflect the theme of the paper. The major improvement is multi-scale INR. The objective itself does not feel like the main novelty.
2. The selection of $w_{low}$ and $w_{high}$ seems to be arbitrary. These are important to the method, but the paper does not explain how they are selected (rules of thumb? dataset-specific tuning? sensitivity ranges?) or how stable the method is to these choices. Since these are presented as key innovations, I think it needs to be much clearer.
3. Some claims are two strong and adding isoPTV comparison in the results would clearly improve the clarity of the presentation.
I would be cautious with wording like “state-of-the-art precision” and “sub-millimeter precision”.  The paper acknowledges that it does not outperform isoPTV, but this comes rather late. It would improve clarity to include isoPTV directly in Table 1 so readers can immediately see where the method stands. Also, for “sub-millimeter precision,” Ref [2] suggests observer error can be close to 1 mm, which makes that claim tricky without careful framing.
Ref [2]: A reference dataset for deformable image registration spatial accuracy evaluation using the COPDgene study archive (https://iopscience.iop.org/article/10.1088/0031-9155/58/9/2861)

**Detailed Comments:**

1. Eq. (10) combines deformations by addition instead of composition. It’s not clear why this choice was made and whether it impacts performance. Some earlier DL work (e.g., LapIRN) uses additive refinement, but most conventional DIR pipelines combine transforms via composition, and many recent DL methods follow that convention. A brief justification (and/or an ablation) would help.
2. The mean TRE of IDIR on COPDgene seems heavily affected by a complete failure on case 6 (outlier). It would be helpful to explain why IDIR fails there, especially since the paper says the single-branch baseline is a controlled variant that differs “only in the loss functions and training protocol.” If the difference is really just training protocol + loss, then the failure mode is important to understand.

**Justification Of Final Rating:**

The authors have addressed my major concerns, and the added results (robustness analysis and regularity quantification) have greatly improved the completeness and quality of the paper. While the method does not outperform isoPTV, this suggests there remains room to further improve the multi-resolution optimization. Overall, the revisions strengthen the paper, and the proposed direction appears promising.

**Justification Of The Preliminary Rating:**

The method is sound, and the improvements look meaningful. I also like the overall direction: bringing a robust classical multi-resolution strategy into INR-based DIR instead of relying on extra architectural tricks. That said, a few key details are currently under-explained (especially selection of $w_{low}$ and $w_{high}$ and the additive deformation combination).

**Questions To Address In The Rebuttal:**

1. Modify the title to clearly reflect the contribution.
2. How are $w_{low}$ and $w_{high}$ selected in practice?
3. Why use addition instead of composition in Eq. 10?
4. Please adjust the SOTA and “sub-millimeter precision” claims, and consider adding isoPTV results directly into Table 1 for better transparency.

---

> ### Author Response · Authors · 2026-01-24
> **Response to Reviewer 6CQ5**
>
> We thank the reviewer for the detailed and insightful assessment. We appreciate the recognition that combining multi-resolution strategies with Implicit Neural Representations (INRs) is a meaningful step forward.
>
> **R.1.1: Title does not reflect the contribution (Multi-scale INR).**
> We agree that the previous title was imprecise. As suggested, we have modified the title to highlight the core contribution and the specific problem domain.
> New Title: *"Robust Multi-Scale Implicit Neural Representations for Large-Deformation Lung Registration"*
>
> **R.1.2: Selection of $\omega$ seems arbitrary.**
> We acknowledge that the rationale for these parameters was under-explained. We chose $\omega_\text{fine} = 30$ following standard SIREN implementations (Wolterink et al., 2022, Sitzmann et al., 2020) and $\omega_\text{coarse}=10$ to bias the coarse branch towards lower spatial frequencies. In response to your comment, we ran a sensitivity analysis (added to the revised mansucript) and found that perfomance remains stable for $\omega_\text{coarse}\in[10,30]$. This indicates that while the lower frequency helps theoretical seperation of motion componentes, the method is robust and does not require precise tuning of this parameter. We have therefore updated the corresponding parts in the manuscript to reflect this.
>
> **R.1.3: Addition instead of composition**:
> We apologize for any confusion regarding this formulation. We model the displacement vector field $u(x)$ (where $\phi(x) = x + u(x)$) rather than the spatial mapping function directly. Consequently, combining two static displacement fields is mathematically equivalent to the addition of their vectors, i.e., $u_\text{total} = u_\text{coarse} + u_\text{fine}$.
>
> **R.1.4: SOTA claims and adding isoPTV to Table 1.**
> We agree that transparency regarding classical benchmarks is essential. We have updated Table 1 to explicitly include a row for isoPTV. Furthermore, we have adjusted the claims of SOTA and sub-millimeter precision in the manuscript to more accurately reflect the comparison with established classical optimization methods.

---

### Author Rebuttal · Authors · 2026-01-24

**Rebuttal:**

We thank the reviewers for their time and their constructive, detailed feedback. We are encouraged that Reviewers 6CQ5 and 1NDy found our approach of integrating multi-scale strategies into INRs to be "meaningful," "practical," and "well-grounded." We have also taken the critical feedback from Reviewer a3gr regarding framing and evaluation rigor very seriously.

In responce to these suggestions, we have strengthened the manuscript with the following revisions:

* **Robustness Analysis (10 Seeds):** We expanded our evaluation to include 10 distinct random seeds for all models to prove stability.
* **Loss Function Ablation:** We added a comprehensive ablation study for the regularization terms.
* **New Experiments Section:** We reorganized the manuscript by adding a dedicated experiments section (Sec. 2.8) to describe the ablation settings.
* **New Visualizations:** We added Figure 3 to visualize the distribution of TREs across random seeds as well as Figure 4 to display more quantitative results.
* **Revised Framing:** We revised the introduction and discussion sections to clarify our specific contributions and address the terminology concerns raised.

We believe these additional experiments and textual revisions directly address the reviewers' concerns, transforming the manuscript into a robust validation of multi-scale INRs for DIR in medical imaging. All changes are highlighted in red in the revised manuscript.

**Supporting Material:**

/attachment/beda658cef454e442fa1ceb30c02368f5593c538.zip

---

### Comment · Area_Chair_1cbb · 2026-01-26
**Start of Discussion Phase**

Thanks to the authors for providing a detailed rebuttal and for engaging constructively with the reviewers’ comments.

I kindly ask all reviewers to carefully read the rebuttal and assess whether the authors’ responses sufficiently address the raised concerns. If aspects remain unclear or require further clarification, please use the official discussion/comments function to ask follow-up questions and engage in discussion with the authors during this phase. Reviewers are also welcome to comment on and respond to the reviews and rebuttal points raised by other reviewers, where relevant.

Please note that by the end of the discussion period, all reviewers are expected to verify and, if appropriate, update their scores. Even if you agree with the authors’ responses but decide not to change your rating, please leave a brief comment indicating that you have read and considered the rebuttal.

---

### Comment · Reviewer_a3gr · 2026-01-27
**Qualitative results and folding ratio**

I would like to thank the authors for their detailed responses and their efforts. I think this led to a huge improvement in the paper quality. However, I still believe the methods' framing remains incremental and that the text could be improved. Regarding the qualitative results, I believe it is quite nice that the authors added them. However, I believe the qualitative results are better if the difference image between the fixed and warped images is added, along with a visualization of the underlying transformation. Finally, I am still missing an indication of the folding ratio for the different methods, or a metric that indicates whether the resulting transformations are plausible.

---

> ### Author Response · Authors · 2026-01-28
> **Response to Reviewer a3gr**
>
> Thank you for your additional feedback. We are pleased that our previous revisions improved the manuscript. We agree that further evidence regarding the plausibility of the estimated transformations strengthens the paper, and we have incorporated the following updates:
>
> 1. **Folding-ratio values:** We have added the folding-ratio ($|J_\phi| \leq 0$) to all ablation models in Table 2. The results demonstrate that the INR-based fields maintain consistent smoothness across configurations, except in explicitly failed cases (dual-branch no multi-scale).
> 2. **Visualizing field smoothness:** We have updated Figure 4 (qualitative results) by overlaying the fixed images with a regular grid and displaying the corresponding forward-warped grid on the moving images. The grid points are color-coded by lateral (out-of-plane) motion magnitude. This visualization provides qualitative evidence of the smoothness and anatomical plausibility of our predicted displacement fields.
>
> All changes are highlighted in **orange** in the revised manuscript.

---

### Comment · Reviewer_6CQ5 · 2026-01-28
**Regularity metrics**

I would like to thank the authors for their response and revision. I also thank my fellow reviewers for raising very constructive comments. The revised paper has improved substantially. In particular, the added robustness analysis is important and adds to the paper’s incremental contribution.

I also agree with reviewer a3gr that the paper would be more complete if it could include deformation regularity metrics (e.g., folding percentage or non-diffeomorphic volume [https://github.com/yihao6/digital_diffeomorphism]) across methods and key ablations. This is relevant because the proposed multiresolution strategy can be plausibly linked to improved deformation regularity. Quantifying this effect would strengthen the technical claims and the overall evaluation.

---

> ### Author Response · Authors · 2026-01-28
> **Response to Reviewer 6CQ5**
>
> Thank you for the additional feedback and for acknowledging the improvements made to the manuscript. As noted in our response to Reviewer a3gr, we have added further evidence concerning the plausibility of the transformations:
>
> 1. **Topological regularity:** Folding-ratio values have been added to the ablation study in Table 2. The results demonstrate that the INR-based fields maintain consistent smoothness across configurations, except in explicitly failed cases (dual-branch no multi-scale).
> 2. **Deformation field visualization:** Figure 4 has been revised to include warped grid point overlays. By visualizing how a regular grid is deformed and color-coding it by lateral displacement, we demonstrate the high degree of regularity and lack of local distortions in our predicted fields.
>
> All changes are highlighted in **orange** in the revised manuscript.

---

### Meta-Review · Area_Chair_1cbb · 2026-02-09

**Recommendation:** Accept (Poster)
**Confidence:** 5

**Metareview:**

Justification:
The reviewers converged on a positive assessment, with two weak accept and one strong accept recommendations. While the contribution is incremental, the paper demonstrates that well-established concepts from classical deformable registration, such as multi-resolution strategies, transfer effectively to INR-based registration and yield consistent performance improvements. The work is technically sound, clearly written, and empirically well supported within its scope, making it a suitable contribution for the conference and best suited for a poster presentation given its limited conceptual novelty.

**Summary:**

The paper proposes an extension of INR-based deformable registration by incorporating a classical multi-resolution strategy. The method employs staged optimization with progressively increasing spatial resolution and a two-branch design that separates large global motion from fine local deformations. Additional loss terms are introduced during INR fitting to improve robustness.

**Strengths**

1.	Effectively integrates a classical multi-resolution strategy into INR-based deformable registration, addressing an underexplored but practically important aspect of INR methods.

2.	Emphasizes simple, well-motivated design choices over architectural complexity, consistent with insights from traditional and learning-based registration.

3.	Proposes a clear and well-designed multi-resolution and dual-branch INR fitting strategy.

4.	Strong ablation study that isolates and validates the impact of the proposed components.

**Remaining weaknesses**

1.	The overall contribution remains somewhat incremental, as the core idea builds on established INR-based registration (e.g., IDIR) and classical multi-resolution strategies; despite improved framing, the novelty may still appear limited to some readers.
2.	Claims regarding the advantages of INRs over classical iterative optimization methods (e.g., grid resolution, optimization landscape) are now better qualified but are still not directly supported by experimental comparisons to non-INR iterative baselines.
3.	The evaluation remains limited in scope, as experiments are conducted on lung CT datasets only, with no validation on other anatomical regions or modalities.

---

### Decision · Program_Chairs · 2026-02-14

Accept (Poster)